# Subcellular Localization of miRNAs and Implications in Cellular Homeostasis

**DOI:** 10.3390/genes12060856

**Published:** 2021-06-02

**Authors:** Minwen Jie, Tong Feng, Wei Huang, Moran Zhang, Yuliang Feng, Hao Jiang, Zhili Wen

**Affiliations:** 1Laboratory for Aging and Cancer Research, National Clinical Research Center for Geriatrics, West China Hospital, Sichuan University, Chengdu 610041, China; minwenjie1996@foxmail.com (M.J.); fengt9@mail2.sysu.edu.cn (T.F.); 2Key Laboratory of Transplant Engineering and Immunology, NHC, West China Hospital, Sichuan University, Chengdu 610041, China; 3Key Laboratory of Gene Engineering of the Ministry of Education, State Key Laboratory of Biocontrol, School of Life Sciences, Sun Yat-sen University, Guangzhou 510275, China; 4Department of Pathology and Laboratory Medicine, University of Cincinnati College of Medicine, Cincinnati, OH 45267, USA; huangwe@ucmail.uc.edu; 5Department of Gastroenterology, The Second Affiliated Hospital of Nanchang University, Nanchang 330006, China; simera1996@outlook.com; 6Botnar Research Centre, Nuffield Department of Orthopaedics, Rheumatology and Musculoskeletal Sciences, Old Road, Headington, Oxford OX3 7LD, UK

**Keywords:** microRNA, subcellular localization, transcriptional regulation, phase separation, condensate

## Abstract

MicroRNAs (miRNAs) are thought to act as post-transcriptional regulators in the cytoplasm by either dampening translation or stimulating degradation of target mRNAs. With the increasing resolution and scope of RNA mapping, recent studies have revealed novel insights into the subcellular localization of miRNAs. Based on miRNA subcellular localization, unconventional functions and mechanisms at the transcriptional and post-transcriptional levels have been identified. This minireview provides an overview of the subcellular localization of miRNAs and the mechanisms by which they regulate transcription and cellular homeostasis in mammals, with a particular focus on the roles of phase-separated biomolecular condensates.

## 1. Introduction

MicroRNAs (miRNAs), a class of evolutionarily conserved endogenous small noncoding RNAs (ncRNAs), contain approximately 20–22 nucleotides. Since their initial discovery, an increasing number of miRNAs have been identified in invertebrates, vertebrates, and plant genomes [1,2,3], including ~2000 miRNAs in the human genome [4]. As the most extensively studied class of ncRNAs, miRNAs play important roles in many cellular biological processes, such as proliferation, differentiation, apoptosis, and stress responses [5,6]. They are key regulators and biomarkers [7,8,9] of human diseases, such as neurological diseases [10], cardiovascular diseases [11], cancer [12,13], and aging [14]. For example, fragile X syndrome-related miR-125b and miR-132 could modify synaptic strength and regulate synaptic structure [15]. Although miRNAs were initially discovered for their role in cell fate and differentiation decisions during organismal development, they are equally crucial in modulating different homeostasis, including endothelial [16], bone [17], and gut [18]. For instance, miR-876-3p regulates glucose homeostasis and insulin sensitivity by targeting the adiponectin system molecules [18].

MiRNA genes could either be located in intragenic regions and share transcriptional regulatory units with host genes [19,20,21] or be found in intergenic regions of the genome with independent *cis*-regulatory elements (CREs) [19,22,23]. MiRNA biogenesis typically involves processing from primary miRNA transcripts (pri-miRNAs) to precursor miRNAs (pre-miRNAs) and mature miRNAs in the nuclei and cytoplasm (Figure 1). Most miRNAs are canonically transcribed as large pri-miRNAs with a 5′-cap and/or poly-A tail by RNA polymerases II or III in nuclei [24]. Pri-miRNAs are long double-stranded RNAs composed of an apical loop, a stem of ~35 base pairs, and two flanking single-stranded nucleotides (Belt and Wedge) at the basal end [25]. Then, pri-miRNAs are further cleaved into pre-miRNAs by a microprocessor complex containing the RNase III enzyme DROSHA and RNA binding protein DiGeorge syndrome critical region 8 (DGCR8) in the nucleus [25,26]. During this cleavage event, the DGCR8 double-stranded RNA binding domain (dsRBD) in the apical half and the DROSHA dsRBD in the basal half form a “double-dsRBD” molecular ruler for an ~35 base pairs stem region [25]. Moreover, pre-miRNAs with a stem-loop structure are transported from the nucleus to the cytoplasm through the exportin/RanGTP complex [27]. Then, after the removal of an apical loop by the RNase III endonuclease DICER and trans-activation-responsive RNA binding protein (TRBP) [28,29], pre-miRNAs are further processed into a miRNA duplex with a phosphorylated 5′ end and hydroxylated 2 nt 3′-overhang [30]. Next, the miRNA duplex is unwound and the guide strand along with Argonaute (AGO) protein form the miRNA-induced silencing complex (miRISC), binding to the target mRNAs to suppress gene expression [31].

MiRNAs were initially thought to regulate gene expression in the cytoplasm [3] negatively. Most studies of the gene regulatory functions of miRNAs have focused on sequence-specific mRNA degradation or 3′ UTR “seed-based” translational repression at the post-transcriptional level in the cytoplasm [6]. Nowadays, accumulating data, including small RNA deep sequencing data, has clarified the subcellular location of miRNAs in the nuclei [32] and revealed various novel functions in cellular homeostasis [33]. By unconventional methods, miRNAs could encode small peptides, interact with splicing machinery to regulate gene transcription, and even directly activate target gene transcription [33]. Recently, a study reported that inflammatory miRNAs, including miR-146a, miR-142-3p, and miR-142-5p, are primarily located on mitochondria-associated ER membranes (MAMs) in human and rat brains [34]. Additionally, they showed a distinct distribution pattern in response to inflammatory injury and cell stress [34]. However, the potential roles of miRNA subcellular localization in modulating cellular homeostasis have not been well characterized.

Currently, some researchers have proposed that miRNAs could contribute to the regulation of cellular homeostasis through phase separation [35,36]. This means that biomacromolecules such as proteins and RNAs are separated into two phases through weak multivalent interaction [37], especially when fast, large-scale gene expression regulation is needed (e.g., during stress response). Moreover, it was reported that membrane-less compartments could form via phase separation [37]. Furthermore, increasing evidence supports the idea that membrane-less subcellular compartments, collectively called biomolecular condensates, enhance biogenesis [35] and the functioning of miRNAs [37]. For example, AGO2 and GW182 (components of miRISC) proteins could form functional phase-separated condensates to regulate miRNA processing [35]. Elsewhere, it was reported that alternative transcription initiation of DGCR8 could skip stem-loop structures in exons, resulting in a soluble microprocessor phase-separated into insoluble aggregation and finally impeding the expression of global miRNA [37]. This global miRNA dosage control is conserved in the HepG2 and K562 cells and human tissues [37]. Phase separation is not an uncommon phenomenon. However, how the phase-separated condensates function on physiological and biochemical activities remains largely unknown. As a newly emerging concept, phase-separated condensates have shown great potential to better understand human disease pathogenesis [38,39], including cancer.

Therefore, this review provides a brief overview of the subcellular localization of miRNAs and the mechanisms by which they regulate transcription and cellular homeostasis in mammals, focusing on the phase-separated biomolecular condensate perspective.

## 2. Nuclear Localization

In the past two decades, the nuclear localization of many miRNAs has been well-established. Mature miR-21 was first detected in isolated HeLa cell nuclei using quantitative northern blotting [40]. Proteins associated with miRNAs, like the glycine tryptophan protein of 182 kDa (GW182) family protein TNRC6A, have been shown to participate in nuclear miRNA functions [41]. Advances in next-generation sequencing technology have identified numerous miRNAs, mainly enriched in the nucleus. For example, the miR-3535 is enriched in the nucleus via a 27.9-fold over the cytoplasm in murine C116 cells [42]. The high abundance of miRNAs in the nucleus suggests their functional roles in transcriptional and post-transcriptional regulation in cellular homeostasis and human diseases. Recently, nuclear miR-30b-5p was reported to target coordinated lysosomal expression and regulation (CLEAR) elements and negatively regulate lysosomal biogenesis and autophagy in a transcription factor EB (TFEB)-dependent manner [43]. Furthermore, nuclear miR-320 up-regulates CD36 expression by directly activating transcription, whereas the inhibition of nuclear miR-320 could rescue cardiac dysfunction in diabetic mice [44,45]. This section has mainly described the molecular mechanisms underlying the effects of nuclear miRNAs during transcription regulation.

### 2.1. Nuclear miRNAs Acting on Cis-Regulatory Elements (CREs)

Recent studies demonstrated that miRNAs located in the nucleus could act on CREs (e.g., promoters and enhancers) to regulate gene expression (activation or suppression) (Figure 2). For example, the nuclear miR-665 binds directly to the *PTEN* promoter and suppresses its transcription, promoting heart failure [46]. Nuclear miR-320 can bind to the promoters of multiple genes (*CEP57*, *SORBS3*, *MEX3A*, *FSCN2*, *PTGIR*, *VPS37D*, etc.), activating their transcription [47]. MiR-320 translocates from the cytosol into the nucleus, binds to the promoter of *CD36* and promotes its transcription, thereby increasing lipid deposition in hepatocytes [48]. Nuclear miR-373 and miR-744 can bind the *CDH1* (*E-cadherin*) and *CCNB1* (*cyclin B1*) promoters, respectively, to induce expression and modulate tumor growth [49]. However, the precise mechanism in which miRNAs up-regulate or down-regulate gene expression by targeting promoters remains unclear. A recent report suggested the potential roles of the interaction of AGO and nuclear miRNAs. For example, the miR-133a/AGO2 complex binds to the *DNMT3B* promoter, stimulating promoter methylation using DNMT3B [50]. The knockdown of AGO2 attenuates the miR-665-mediated PTEN up-regulation in H9c2 cells [46]. In addition, the expression of nuclear AGO rescues CD36 overexpression caused by the direct binding of miR-320 to the *CD36* promoter [44]. However, based on the DNA-RNA hybrid, some researchers have argued that AGO could not be necessary for miRNA–promoter interactions [51,52]. For example, Zhang et al. reported that let-7i directly targeted the TATA-box region of the IL-2 promoter [52]. Thus, miRNAs directly interact with the DNA promoter, providing another possible mechanism independent from AGO for miRNA–promoter interactions. Hence, the precise roles of AGO in miRNA-mediated transcription require further investigation.

Apart from promoter binding, nuclear miRNAs could also act on enhancers to activate transcription (Figure 2B). Enhancer triggering nuclear activating miRNAs (ET-NamiRNAs), bound on active enhancers marked by histone H3K27ac, P300/CBP, and DNase I high-sensitivity loci, were first described in 2017 [53]. ET-NamiRNAs could promote the transcription of homologous genes distributed across the genome. This process could regulate cell reprogramming and induce tumorigenesis [44,54]. As validated through the luciferase assay, miR-26a-1 activates enhancers to up-regulate neighboring genes, such as *ITGA9* and *VILL* [53]. Moreover, miR-3179 and miR-3180 are located in the same chromosomal region next to *ABCC6* and *PKD1P1*. While the miR-3179 with enhancer activity up-regulated the neighboring genes *ABCC6* and *PKD1P1*, the miR-3180 without the enhancer activity failed to activate these genes [53]. Furthermore, miR-24-1 promoted the expression of neighboring *FBP1* and *FANC* genes. The overexpression of this miRNA depended on AGO2, which was functioned by enriching the RNA Pol II and altering the chromatin accessibility of enhancers and promoters [53]. Furthermore, super-enhancers (SEs) and transcription factors (TFs) exert cell type-specific effects on miRNA processing. The SEs recruit DROSHA/DGCR8 and promote pri-miRNA processing, networking with miRNAs in at least 18 cancer cells [55]. Generally, miRNAs interact with enhancers to modulate gene expression and cellular homeostasis. However, the molecular mechanism underlying miRNA–enhancer interactions and functions remain obscure.

Condensates formed by phase separation influence transcription. For example, the coactivators MED1 and BRD4 form nuclear puncta at SEs in mouse embryonic stem cells. Condensates formed by the intrinsically disordered region of MED1 (MED1-IDR) attract BRD4 and RNA Pol II and squelch in vitro transcription with nuclear extracts [56]. Interestingly, the miRNA-related protein GW182 undergoes phase separation in GFP-GW182-overexpressing HEK293 cells [35]. Moreover, AGO2 and GW182 form phase-separated condensates through in vivo and in vitro setups, as confirmed using fluorescence recovery after photobleaching (FRAP) and fusion assays [35]. Furthermore, the AGO2-GW182 condensate recruited miRISC-associated proteins and promoted the deadenylation of target RNAs [35]. Based on AGO2 and GW182 nuclear locations, miRNA-associated proteins (e.g., AGO2) could modulate transcription through phase separation on CREs. Young and colleagues reported that nuclear condensates formed using TFs and coactivators, including MED1, could include or exclude small molecules, providing a basis for developing therapies against chemoresistant cancer [57]. Based on these previous findings, we have proposed a model for the effects of nuclear miRNAs on CREs (Figure 3). First, nuclear miRNA-associated miRISC components, such as AGO2 and GW182, undergo phase transitions to form condensates at targeted CREs. On the one hand, specific RNA molecules (e.g., lncRNAs and miRNAs) and proteins (e.g., TFs, RNA Pol II, and chromatin remodeling complexes) could be recruited to facilitate or impede transcription by promoting transcription initiation and elongation or remodeling chromatin. On the other hand, the condensates could exclude the transcription activation machinery that down-regulates gene expression.

MiRNAs acting on CREs provide new targets and strategies for preventing and treating of cancer and other human diseases. This emphasizes the importance of further investigations of the interactions between miRNAs and CREs and their biological and physiological functions in cellular homeostasis and human diseases.

### 2.2. Other Potential Pathways for Nuclear miRNAs to Regulate Transcription

Beyond direct interactions with CREs, other potential pathways, including the regulation of RNAs, peptide activity, and chromatin remodeling [58], for transcriptional regulation through nuclear miRNAs, have emerged.

The ability of miRNAs to silence the expression of TFs has been established. For example, miR-9 suppresses the expression of peroxisome proliferator-activated receptor δ (PPAR-δ), which binds to the 3′UTR of *PPAR-δ* mRNA, as validated using a luciferase assay in human monocytes [59]. Apart from the cytoplasm, miRNAs also exert post-transcriptional gene silencing in the nucleus. Research has reported that miR-9 directly targets metastasis-associated lung adenocarcinoma transcript 1 in the nucleus and degrades it in an AGO2-dependent manner [60]. Nuclear miR-122 has been shown to prevent the maturation of pri-miR-21 by directly binding on it, thus blocking the recognition region of the DROSHA-DGCR8 microprocessor, promoting liver cell apoptosis [61]. Similarly, miRNA-709 impedes miRNA-15a/16-1 biogenesis to regulate apoptosis in mouse cells [62]. Based on different RNA-mediated transcription studies [63,64], we speculate that nuclear miRNAs could indirectly modulate transcription by degrading genes encoding TFs and preventing the maturation of ncRNAs (Figure 2C).

Interestingly, pri-miRNAs produce peptides, called miRNA-encoded peptides (miPEPs), which upregulate the transcription of these pri-miRNAs. For example, pri-miR-171b in *Medicago truncatula* and pri-miR-165a in *Arabidopsis thaliana* were first described to function as template-encoding peptides of nine and 18 amino acids, respectively [65]. Moreover, these two miPEPs participate in root development by up-regulating their pri-miRNAs, resulting in mature miRNA accumulation and the down-regulation of target genes [65]. Furthermore, five other miPEPs in *M. truncatula* and *A. thaliana* [65] and two other miPEPs (miPEP-200a and miPEP-200b) in prostate cancer cells [66] have been identified. A miR-34a pri-miRNA encoded miPEP133 was identified in many human tissues, including colon, stomach, ovary, uterus, and pharynx and during regulated apoptosis, invasion, and migration of cancer cells [54]. Collectively, these data suggested that miRNAs could encode short peptides miPEPs as new factors during transcription regulation (Figure 2D).

Increasing evidence suggests that RNAs contribute to the organization of the 3D chromatin structure [64]. For example, nascent RNA-induced R-loops are enriched with TFs and chromatin remodeling factors in mouse embryonic stem cells to impair differentiation [67]. Moreover, the miRNA lethal 7-mediated multicomponent RNA–protein complex (MiCEE), including polycomb repressive complex 2 (PRC2), induces heterochromatin and silences transcription [58]. Interestingly, mutation of RNA-binding regions in CTCF (a critical architectural protein) drastically disrupted chromatin looping and impeded gene expression [68], suggesting that the chromatin-associated RNAs (e.g., nuclear miRNA) could be involved in their 3D genome organization (Figure 3C). Further, a number of droplet-like condensates are associated with RNAs, partly because most proteins involved in phase separation are RNA-binding proteins [63,69,70]. Therefore, it is imperative to use phase separation or other mechanisms in the future to explore whether nuclear miRNA modulates chromatin topology.

## 3. Membrane Compartments

Membrane compartments are functional subcellular organelles surrounded by membranes that mainly consist of lipids and proteins. By participating in the material exchange, and energy and information transfer process, membrane compartments play essential roles in promoting cell survival, growth, division, and differentiation, thereby maintaining homeostasis of the intracellular microenvironment. Notably, the most crucial membrane compartments associated with miRNAs are mitochondria and endoplasmic reticulum (ER) (Figure 4).

### 3.1. Mitochondria

Mitochondria are double membrane-bound cellular compartments in eukaryotic cells with a central role in cellular homeostasis and human diseases. As the primary sites for redox reactions and ATP production, mitochondria have a genome similar to bacterial chromosomes. The human mitochondria DNA (mtDNA), with ~16,000 base pairs length, encodes only 13 proteins (~1% of all proteins) [71]. However, the mtDNA encodes abundant small non-coding RNAs [72] (e.g., the majority of mitochondrial tRNAs are encoded by mtDNA) [73]. In the purified human 143B cell mitochondria, several small RNAs mapped to mtDNA have been identified using small RNA sequencing, without the DICER-dependent characteristic of nuclear tRNAs [74]. Moreover, through Sanger sequencing, small RNA RT-PCR, and PCR amplification of mouse and human small RNA libraries, Ro et al. demonstrated that the mitochondrial genome encodes small RNAs (mitosRNAs). These included miRNAs with 5′ phosphate termini different from the 5′ termini of RNA turnover products [72]. However, due to the absence of miRNA processing proteins, such as DROSHA, in the mitochondria, the processing and function of miRNAs transcribed from mtDNA have not been well understood.

Additionally, nuclear miRNAs in different species have been shown to localize and function in the mitochondria. For instance, a qPCR analysis verified that the miR-4485 levels in the mitochondria are more than two-fold higher than those in whole HEK293 cells, using U6 snRNA as the endogenous control [75]. Additionally, the levels of precursor miR-4485 in the cytosol were approximately six times higher than those of precursor miR-4485 in the mitochondria [75]. Furthermore, in HEK293 cells, the miR-4485 levels were reduced by the deleting mtDNA using bromide treatment, and mitochondrial miR-4485 was down-regulated when cytoplasmic translocation was inhibited using Leptomycin B [75]. Nuclear miR-181c is translocated to mitochondria in cardiac myocytes [76]. Although possible models explaining the miRNAs translocation into the mitochondria have been proposed [77] (Table 1), the specific mechanism and biological significance of this translocation are yet to be established.

Mitochondrial miRNAs (MitomiRs) play vital roles in the regulating mitochondrial function in human diseases, such as cancer. For instance, Fan et al. reported that mitomiR-2392 disrupts the metabolic balance between downregulated oxidative phosphorylation (OXPHOS) and upregulated glycolysis in cisplatin-resistant tongue squamous cell carcinoma cells and tumors [84]. MitomiRs contribute to diseases mainly through mitochondria-mediated processes, such as apoptosis and energy metabolism. For example, mitochondrial miR-762 was reported to regulate redox reactions and ATP production in the mitochondria and to promote apoptosis via NADH dehydrogenase subunit 2 (ND2) in cardiomyocytes [85]. Similarly, mitochondria-targeted miR-4485 regulates mitochondrial metabolism and apoptosis in breast carcinoma cells [75]. Thus, future studies on specific mechanisms underlying mitomiR distribution and functions could identify new drug delivery routes and mitochondrial biomarkers to treat mitochondria-based human diseases.

### 3.2. Endoplasmic Reticulum

ERs, with numerous metabolic functions, have crucial roles in protein assembly and secretory pathways. In addition, the ER has vital roles in Ca^2+^ storage and lipid and sterol biosynthesis. Rough ER is enriched with membrane-attached ribosomes, which are the primary sites for translating different proteins such as membrane-spanning, secreted, and soluble proteins [86]. Of note, AGO1–miRNA complexes were found to co-sediment with ribosomes in both iodixanol gradient and sucrose gradient centrifugation assays [87]. Moreover, using the parallel analysis of RNA ends (PARE) technique, Yang et al. reported that miRNA-guided target cleavage events were pervasive in ribosome-associated mRNAs of maize and rice [88]. For example, the PARE assay suggested that miR-2118 and miR-2275 levels are higher in membrane-bound polysomes than in total polysomes and cleaved 21PHAS and 24PHAS precursors in tassels and panicles [88]. Taken together, ER-membrane-attached ribosomes could enrich miRNA and provide a place for miRNA-guided target cleavage.

Recently, miRNAs have been implicated in regulating ER stress, in which the disturbance of protein translation, modification, and degradation was caused by extracellular and/or intracellular stress. For example, ER stress in HUVECs caused by tunicamycin treatment resulted in the up-regulation of miR-204, and, conversely, treatment with a miR-204 inhibitor could prevent the ER stress [89]. The specific mechanism by which miRNAs contribute to the ER stress response is complicated and studies have largely primarily focused on protein folding, modification and the unfolded protein response (UPR). An in-silico analysis and luciferase assay identified miR-224 and miR-520c as suppressors of tumor suppressor candidate 3 (TUSC3), related to tumor procession [90]. The knockdown of TUSC3 in orthotopic xenograft mice promoted lung cancer metastasis, which was rescued through anti-miR-224/520c miRNAs treatment [90]. Moreover, recent study suggested that the ER stress-induced UPR sensor activating transcription factor 6 alpha (ATF6 alpha) influenced UPR activation under ER stress, triggered by a TUSC3 deficiency [90]. This was not the case in protein kinase R-like endoplasmic reticulum kinase (PERK) and inositol requiring enzyme 1 alpha (IRE1 alpha). Furthermore, the activation of ATF6 depended on the Cys-X-X-Cys motif, in which TUSC3 promoted the N-linked glycosylation of proteins [90]. Taken together, miR-224 and miR-520c suppressed TUSC3 expression to induce ER stress and impeded the glycosylation of ATF6α to block ER stress-induced UPR. More recently, it has been reported that ER stress induces the expression of miR-26a. By directly binding to eukaryotic initiation factor 2 alpha (eIF2 alpha), miR-26a moderates ER stress and lipid accumulation in non-alcoholic fatty liver disease [91]. The overexpression of miR-204 down-regulates Sirtuin1 and promotes ER stress, along with the accumulation of abnormally acetylated proteins [89].

Additionally, ER-related miRNAs play a crucial role in cancer proliferation, metastasis and apoptosis. For example, the overexpression of miR-328-3p, targeting ER metalloprotease 1 (ERMP1), prevented liver cancer cell proliferation and invasion in female nude mice by reducing phosphorylation [92]. RT-qPCR analyses of biopsy specimens and cell lines verified that miR-370 is upregulated in gastric cancer, and Circ_002117 inhibits miR-370 to stimulate ER stress-induced apoptosis [93]. Moreover, ER stress-induced exosomal miRNAs also contribute to the pathogenesis of cancer. For example, ER stress increased the release of exosomal miR-23a-3p from hepatocellular carcinoma (HCC) cells, promoting HCC genesis via the upregulation of PD-L1 in macrophages [94]. Furthermore, exosomal miR-27a-3p induced through ER stress has been found to activate macrophages by targeting MAGI2 to upregulate PD-L1. The activated macrophages inhibit CD8+ T cells from promoting immune escape in breast cancer [95].

In summary, miRNAs play significant roles in promoting the biochemical function of the ER, including RNA cleavage, protein modification and the UPR. Given the diverse roles of the ER in maintaining homeostasis, it is reasonable to assume that ER stress-induced miRNAs are considerably critical during environmental stress responses.

### 3.3. Endosomal Trafficking

miRNAs are found in the endosomal trafficking pathways, including GW-bodies, multivesicular bodies (MVBs), and endosomes [96,97,98]. Lee et al. suggested that MVBs enhanced the miRNA-guided silencing by promoting recycling of miRISC and contributes to the recruitment of miRNAs to miRISC [96]. Either blocking the formation of MVBs or promoting MVBs to form lysosomes could impair silencing mediated by miRNAs [96]. Except for the recycling model, the cellular trafficking system provides sufficient space for a more distal regulation of miRNA-related silencing. Corradi et al. proposed that pre-miRNAs could be trafficked to distal axons with the help of lysosomes/late endosomes to be processed into mature miRNAs in the distal axon to repress mRNA translation [98]. Collectively, the endosomal trafficking pathway could control the distribution and function of miRNAs.

## 4. Membrane-Less Compartments

In addition to membrane compartments, many membrane-less compartments are formed through phase separation and are involved in cellular homeostasis. Cells have evolved elaborate cellular stress response mechanisms to maintain homeostasis and protect them against stress, including cell repair and cell death [99]. These responses require massive and rapid gene expression regulation at the transcriptional and/or post-transcriptional level. Cells must activate complex response mechanisms over a short time to fight against extracellular stress. Unfortunately, such a “battle” itself is a burden to cells and is likely to cause cell damage. Therefore, it is a challenge to optimize extracellular and intracellular processes while minimizing costs in cells. Accumulating evidence has demonstrated that membrane-less subcellular compartments associated with miRNAs are efficient and convenient tools for maintaining homeostasis.

In the cytoplasm, stress granules (SGs) are among the most extensively studied miRNA-associated membrane-less compartments. SGs, mainly comprising mRNAs and RNA-binding proteins, are cytoplasmic membrane-less compartments regulating mRNA translation and defend against cellular stress to maintain homeostasis [100]. Moreover, SGs are formed through phase separation [101]. Recently, interactions between SGs and miRNAs have been identified. As determined using mass spectrometry, interactions between HSP90 and the co-chaperone p23 with DICER as well as interactions between SG-associated proteins, such as AGO2, and/or DICER increased under stress [102]. Moreover, the stress-induced interactions between SGs and miRNA-related proteins (AGO2 and DICER) could regulate miRNA biogenesis by attenuating DICER catalytic activity [103]. Conversely, in acute ischemic stroke rats model of middle cerebral artery occlusion (MCAO), miR-335 promoted SG formation by downregulating ROCK2 [103]. Further, miR-335 directly targeted the ROCK2 3′ UTR to inhibit apoptosis in PC12 cells [104]. Interestingly, in ischemic cortices of MCAO rats, METTL3, m6A, and miR-335 were up-regulated at 0 h after reperfusion but significantly down-regulated at 24 h after reperfusion [104]. This was similar in PC12 cells treated with oxygen-glucose deprivation/reperfusion (OGD/R) stimulation as an in vitro model of acute ischemic stroke [104]. Furthermore, METTL3-mediated m6A modifications of pri-miR-335 promoted the mature miR-335 expression, enhancing SG formation and attenuating apoptosis in OGD/R stimulated PC12 cells. DCP1A and GW182 were reported to enhance miRNA regulation of targets by inducing the formation of processing bodies and GW bodies [105]. Overall, miRNA interactions with components of cytoplasmic membrane-less compartments could be involved in cellular homeostasis. Conversely, cytoplasmic membrane-less compartments induced by environmental stress could regulate miRNA biogenesis and functions in human diseases.

Processing bodies (PBs) are other prominent cytoplasmic membrane-less compartments, mainly consisting of mRNAs, mRNA cleavage enzymes, and RNA transport factors [106]. In response to stress, PBs interact with SGs and increase their size and number [99,106]. In addition, PBs could intake complex phosphorylated AGO2 [78] and translocate them into mitochondria. More recently, dicing bodies in plants have been regarded as protein condensates formed using phase transitions. During an in vitro setup, SERRATE forms liquid-like droplets to recruit DCL1, HYL1, and pri/pre-miRNAs, promoting the processing activity, maturation, and export of miRNAs [107].

Except for well-known miRNA-related membrane-less compartments, various RNA-related subcellular biomolecular condensates are crucial in resisting environmental disturbances. For example, the SG marker poly(A)-binding protein (PAB1) forms condensates to sense thermal or pH stress and maintains yeast fitness by promoting mRNA translation [108]. A study reported that condensates formed by the translation termination factor Sup35 could rescue the catalytic C domain to preserve the rapid recovery of yeast by reinitiating mRNA translation [109]. In addition to RNA translation, some biomolecular condensates modulate transcription and RNA splicing. Surprisingly, when phosphorylated by CDK7 and CDK9 at the C-terminal domain, RNA Pol II promotes the switch from condensates involved in transcription initiation to condensates related to RNA splicing [110]. Moreover, Henninger et al. proposed an RNA-mediated no-equilibrium feedback model of transcriptional regulation. The model suggested that small amounts of RNA promoted transcription but a large amount of RNA impeded the process [63]. This research shows the possibilities of miRNA-related membrane-less compartments in maintaining cellular homeostasis.

In conclusion, membrane-less compartments and biomolecular condensates could regulate miRNA biogenesis. Further, in cooperation with membrane-less compartments and biomolecular condensates, miRNAs could modulate homeostasis for many biological and physiological processes, including transcription and apoptosis. However, the distribution of miRNAs to these compartments and the specific mechanisms underlying the regulatory effects of membrane-less compartment-related miRNAs on biochemical reactions and biological processes are largely unidentified.

## 5. Conclusions and Perspectives

Since the discovery of miRNAs, their subcellular localization has been largely ignored owing to technical limitations. Although some technical difficulties remain in isolating pure subcellular fractions and in precise immunofluorescence-based localization analyses, accumulating research has provided a broad view of the subcellular localization of miRNAs and novel miRNA functions, especially in nuclei. By studying the mechanisms by which nuclear miRNAs contribute to transcriptional regulation, we encountered a highly complicated network of miRNA–DNA–protein interactions. By directly binding to promotors or enhancers, miRNAs can either promote or inhibit gene expression; however, the specific underling mechanism and molecular pathways are not well-established. We hypothesize that the concept of phase separation may provide a framework for transcriptional regulation by miRNAs. First, the miRNA-related proteins AGO2 and GW182 can form functional condensates [35] and chromatin remodeling factors SWI2/SNF2 can impede pri-miRNA processing [111], suggesting that biomolecular condensates regulate miRNA processing and functions. Moreover, RNA-mediated transcriptional condensates have been verified to modulate transcription initiation and elongation and alter splicing [110]. Furthermore, chromatin-associated RNAs have been implicated in the organization of the 3D genomic structure [63,64], which may partially depend on transcriptional condensates. Taken together, it is likely that miRNAs regulate transcription via phase transitions. Given the fact that bio-condensates are pervasive in cancer cells, strategies related to miRNA-mediated bio-condensates may provide a promising new direction for cancer prevention and therapy.

MitomiRs and ER-located miRNAs may be necessary for the modulation of specific biological processes and tumorigenesis. In addition, miRNAs have been detected in cytoplasmic membrane-less compartments, such as SGs, and these miRNAs are of great significance for cell fitness. However, the organization, trafficking and functions of miRNAs located in different subcellular compartments are largely unknown. Thus, these topics should be further addressed in the future, which could provide novel insights into the development of miRNA-based therapies for human diseases, such as cancer.

## Figures and Tables

**Figure 1 genes-12-00856-f001:**
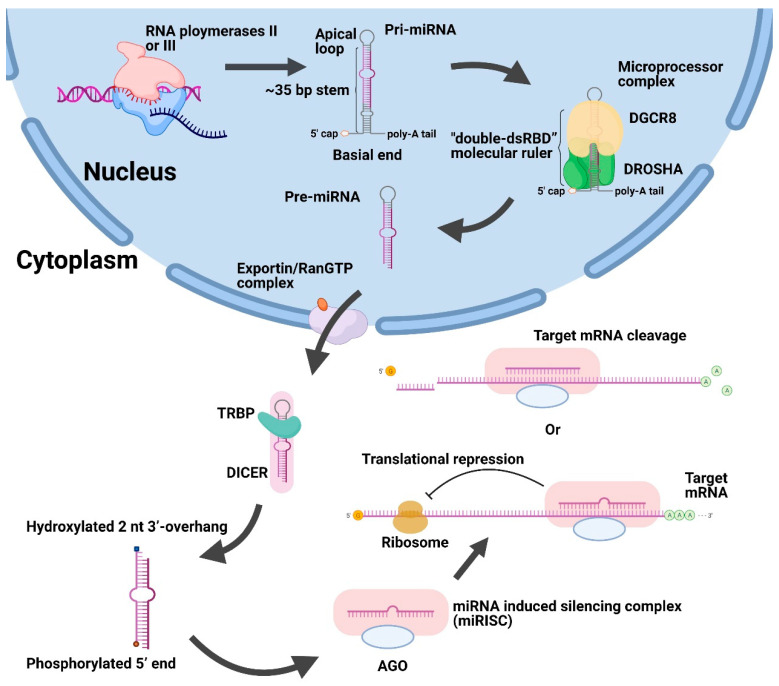
Schematic illustration of canonical biogenesis and function of miRNAs. First, large pri-miRNAs are transcribed by RNA polymerases II or III in the nucleus and then converted into pre-miRNAs after cleavage using the DROSHA/DGCR8 microprocessor complex. Afterwards, the exportin/RanGTP complex exported pre-miRNAs to the cytoplasm. Then, the DICER and TRBP removed the apical loop of pre-miRNAs and unwound the miRNA duplex. After the formation of miRISC, it could suppress the genes by repressing translation or promoting cleavage of target mRNA. Abbreviations: Pri-miRNAs, primary miRNA transcripts. Pre-miRNAs, precursor miRNAs. DGCR8, DiGeorge syndrome critical region 8. TRBP, trans-activation-responsive RNA binding protein. AGO, Argonaute. miRISC, miRNA-induced silencing complex.

**Figure 2 genes-12-00856-f002:**
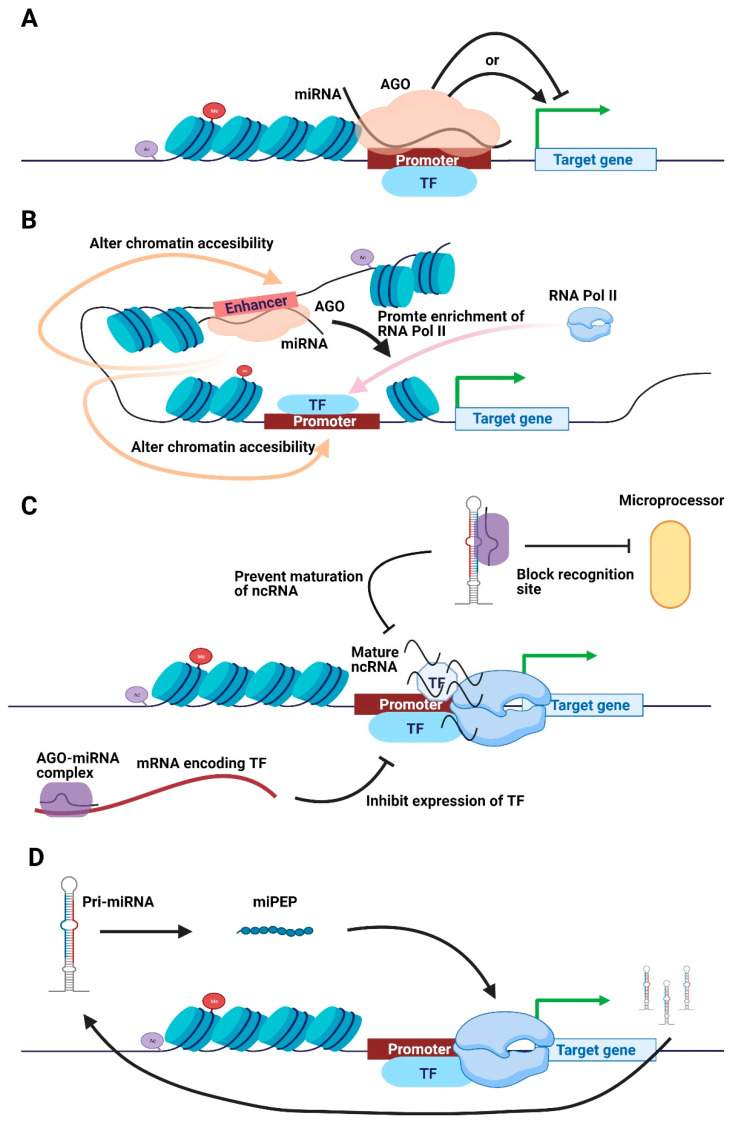
Schematic illustration of transcriptional regulatory mechanisms of nuclear miRNAs. (**A**) Nuclear miRNA directly binds onto the promotor to promote or impede transcription in an AGO-dependent manner. (**B**) Nuclear miRNA and AGO directly bind to the enhancer to enrich RNA Pol II or alter chromatin accessibility, promoting transcription. (**C**) MiRNA inhibits TF expression to suppress transcription, whereas the AGO-miRNA complex blocks the maturation of ncRNA to impede transcription. (**D**) Pri-miRNA encodes short peptides to simulate transcription of responding pri-miRNA. Abbreviations: AGO, Argonaute. TF, transcription factor. miPEP, miRNA-encoded peptide.

**Figure 3 genes-12-00856-f003:**
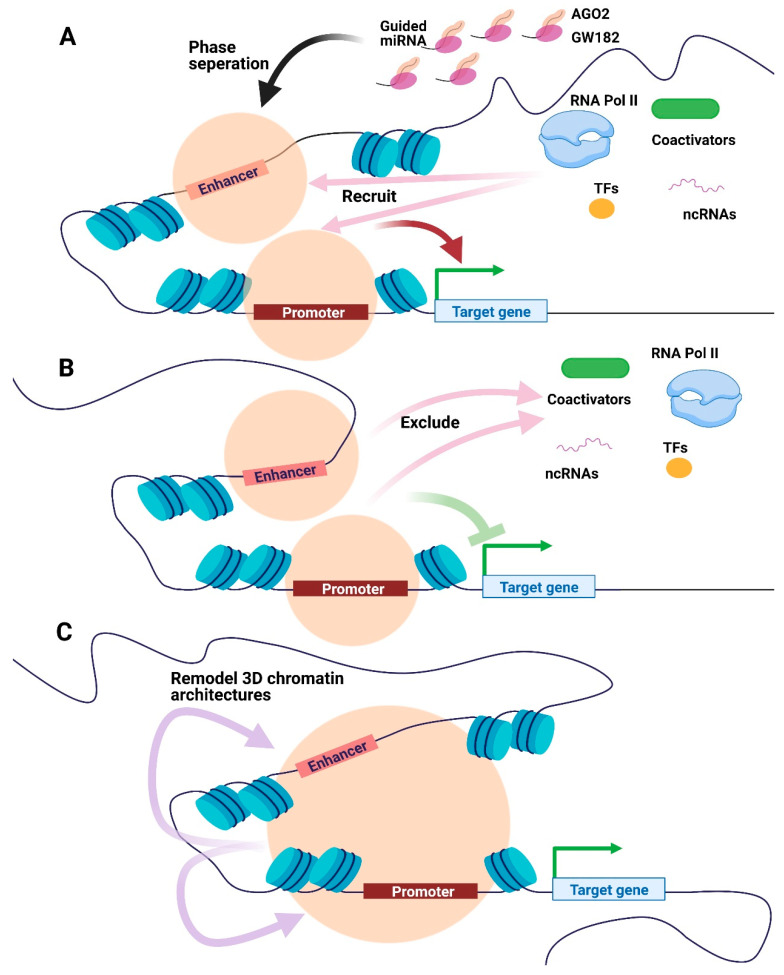
Potential models of miRNA-mediated condensates in transcriptional regulation. (**A**) AGO2, GW182, and guided miRNA could be phase-separated at the promoter or enhancer regions to recruit transcriptional apparatus, including RNA Pol II, TFs, coactivators, and ncRNAs, promoting transcription. (**B**) Condensates consist of AGO2, GW182 and guided miRNA may exclude transcriptional biomolecules, hence impeding transcription. (**C**) The miRNA-mediated condensates could remodel 3D chromatin architectures to regulate transcription. Orange circles represent condensates formed by phase separation. Abbreviations: AGO, Argonaute. RNA Pol II, RNA polymerase II. TFs, transcription factors. ncRNAs, non-coding RNAs.

**Figure 4 genes-12-00856-f004:**
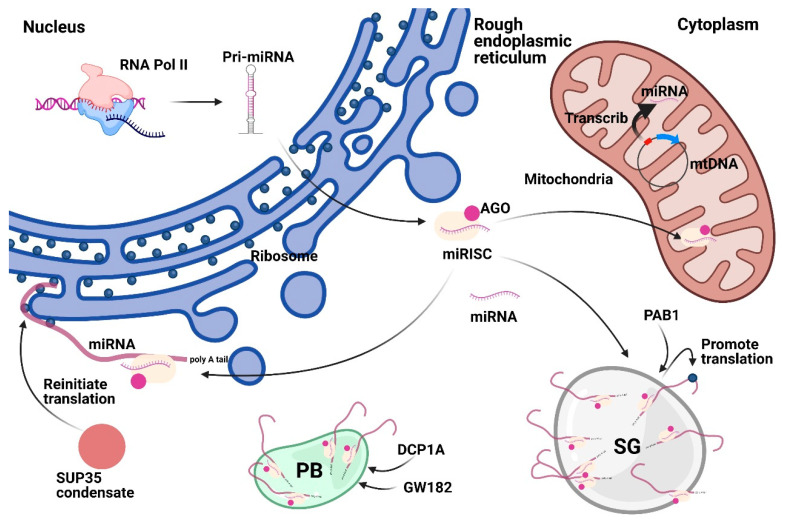
Subcellular localization of Cytoplasmic miRNAs. Cytoplasmic miRNAs could localize in membrane compartments (mitochondria and ER) or membrane-less compartments (SG and PB). In rough ER, miRNA could regulate mRNA translation. Mitochondrion-derived MitomiRs could be encoded from mtDNA or be translocated from the cytoplasm. SG marker PAB1 could promote translation. GW182 and DCP1A could induce PB formation to regulate the function of miRNA. SUP35 could form condensate to reinitiate translation. Abbreviations: AGO, Argonaute. ER, endoplasmic reticulum. SG, stress granule. PB, processing body. MitomiRs, Mitochondrial miRNAs. mtDNA, mitochondria DNA. PAB1, poly(A)-binding protein 1. GW182, glycine tryptophan protein of 182 kDa. DCP1A, decapping mRNA 1A.

**Table 1 genes-12-00856-t001:** Possible mechanism of miRNAs translocation into mitochondria.

Candidated miRNA	Potential Mechanism	Reference(s)
let-7a	Posttranslational modification of AGO2 promotes the AGO2 and miRNA complex intake into P-body, which could translocate them into mitochondria.	[78,79]
	Without AGO2, miRNAs may translocate separately into mitochondria through SAM50 and/or TOM.	[73,77,80]
miR-1	AGO2 and miRNA complex from miRISC is transported into mitochondria after changes of GW182 and/or HDAC4.	[81]
mitomiR-378	Polynucleotide phosphorylase (PNPase) helps import the AGO2 and miRNA complexes into the mitochondria.	[23,82,83]

Abbreviations. AGO, Argonaute. SAM50, sorting assembly machinery 50 kDa subunit. TOM20, Mitochondrial import receptor subunit TOM20. GW182, glycine tryptophan protein of 182 kDa.

## Data Availability

Not applicable.

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
