# Peer review of "Subcellular Localization of miRNAs and Implications in Cellular Homeostasis"

_genes, 2021, doi:10.3390/genes12060856_

Round 1

Reviewer 1 Report

I have read with interest the manuscript by Jie et al. Authors describe the functions of miRNAs in the setting of various cellular contexts. The review is well organized; sometimes, it would benefit from English proof-reading.  Authors shall also provide some basic introduction and explanation of some concepts to the less skilled readers - specific comments can be found below.

Major concerns:

1) Authors are talking about "phase separation," "phase transition," and "biomolecular condensates". It would improve the review's quality if they could provide some introduction to the topic within the introduction section.

Minor concerns:

1) In all figures, please add the heading "Abbreviations" before starting the abbreviations list.

2) Section 2.1, page 4, lines 129-130: Authors state that "some study argued that AGOs are not necessary" - could authors comment on which mechanisms do they propose?

3) Authors state that "miRNAs may indirectly modulate transcription by degrading GENES" - please rephrase, as I cannot imagine how genes may be degraded.

4) Page 5, line 204: Authors state, "Collectively, these data suggested that miPEPs might be a new player in transcription regulation by encoding short peptides (Figure 2D)" Please, rephrase - miPEPs shall be miRNAs, I suppose.

5) Page 9, line 275: Authors state: "Althought possible models of miRNAs translocation into mitochondria have been proposed" - could authors provide some brief list of such proposed examples?

6) Page 11, line 371: Authors are citing the work working with "acute ischemic stroke" - could you please specify whether this was a murine model or a human study?

7) Page 11, lines 386-389: Please rephrase/split the following sentence as it does not make sense: "Moreover,Henningeret al. proposed an RNA-mediated no-equilibrium feedback model of transcriptional regulation, in which small amounts of RNA promoted but large amount of RNA impeded the process"

8) Moderate English proof-reading shall be performed.

Typos:

Abstract, page 1, line 21: "MicroRNAs (miRNAs) ARE traditionally

Abstract, page 1, line 22: Please edit: "With the increasing resolution and scope of RNA mapping, recent studies have provided novel insights into the subcellular localization of miRNAs. And their functionalities. Furthermore, Depending on the miRNA localization, unconventional functions, and mechanisms..."

Introduction, page 1, line 35: erase "canonically."

Introduction, page 1, line 36: please insert "increasing NUMBER OF miRNAs..."

Introduction, page 1, line 38: please insert "As the most extensively studied CLASS OF ncRNAs..."

Introduction, page 1-2, lines 42 - 46: please switch the order of the sentences and insert: "MiRNAs GENES may be located in intragenic regions and share transcriptional regulatory units with host genes or can be found in intergenic regions of the genome with independent cis-regulatory elements (CREs). MiRNA biogenesis typically involves processing from primary miRNA transcripts (pri-miRNAs) to precursor miRNAs (pre-miRNAs) and mature miRNAs in the nuclei and cytoplasm (Figure 1). "

Introduction, figure 1 caption: change "after cleaved" to "after CLEAVAGE."

Introduction, page 3, line 85: change "transcriptome" to "transcriptomic."

Introduction, page 3, line 87: change "By unconventional methods" to "Unconventionally."

Section 2, page 3, line 101: change "extensive miRNAs" to "NUMEROUS miRNAs."

Section 2.1, page 3, line 116: there should be capital "N" in "Nuclear miR-320."

Section 2.2 Page 5, line 187: please edit to "pri-miR-21 by direct binding and thus."

Figure 2 Caption: capital "N" shall be in "(B) Nuclear miRNAs". Also,AGO shall be in the list of abbreviations.

Page 10, line 321: ATF6 ALPHA

Reviewer 2 Report

In this review article, the authors discussed an interesting concept of miRNA localization in subcellular compartments and its implication in cellular homeostasis. Some parts of the manuscript are well written especially the nuclear localization and its role in the regulation of transcription. However, the overall manuscript is difficult to follow and the take-home message is unclear.

General Comment:

-The author used the broad term “Cellular Homeostasis” in the title, however, the discussion was mostly in the field of cancer. The discussion should be expanded to other pathological conditions.

-It would be beneficial to be precise and detail in several contexts as indicated below.

-A recent study suggests that several inflammatory miRNAs are enriched in the mitochondria and the Mitochondrial Associated ER Membrane (MAM) and may play a role in the inflammatory response following traumatic brain injury. (Wang et al., 2020; DOI: https://doi.org/10.1007/s12035-020-01937-y). Including this novel finding in the review would be informative to the readers.  

Comments:

Introduction:

----Line 36:  need to change--- “increasing miRNAs”

----Line 41: need to add more reference for “biomarker”

----Line 45: need to add a reference for “transcriptional regulatory units with host gene”

----Figure 1: In the legends of the figure there are three sections, first section: main title, second section: main text, and third section: spell out abbreviations

I would suggest highlighting its individual parts with clear numbering or in different formatting and also links in the text.

----Line 73-84: This paragraph doesn’t flow well. the authors should discuss the potential roles of miRNA subcellular localization.

Section 2: Nuclear localization:

----Line 105: “nuclear miR-30b” need to added since miR-30b alone does not reflect it's nuclear miRNA identity in this line.

Section 3.1: Mitochondrial localization

----Line 279: The statements in the last paragraph is misleading. According to the cited reference for “ miR-23a and miR-23b”, these miRNAs did not show as MitomiRs.  

Section 3.1: ER

---- Line 306:  authors should provide sufficient information and related references in their conclusions.

Section 4: Membrane-less compartments:

-----No information and reference presented for Processing body (PB).

----line 377-391: The author discussed the role of biomolecular condensates in miRNA biogenesis. The paragraph does not adhere well with the section title. Either remove it or revise it.

----Figure 4: Need change as suggested in figure 1 above.

Reviewer 3 Report

In this manuscript, Jie and colleagues review the recent advances on miRNA subcellular localization in the cellular context of homeostasis. They mostly emphasize on the nuclear roles of miRNAs and present recent evidence suggesting that they modulate transcription. They then go on to discuss miRNAs roles in association with organelles (mitochondria and ER) and finally in membrane-less compartments formed by phase separation. This review is overall well written, well illustrated, focused and interesting to readers with a specific interest on novel roles of miRNAs. It has however some major omissions, which, if addressed, would significantly strengthen the manuscript. 

General comments: 

1. Historically, subcellular RNA localization is associated with cytoplasmic distribution in drosophila oocytes and neurons (synapses, axons). It is therefore surprising that the authors do not mention at all these subcellular compartments in their review, especially as the first compartmentalized functions of miRNAs were described there (in synapses) and local roles of miRNAs in axons is a very active field of study. 

2. Also to my surprise, the authors do not mention at all miRNAs associated to endosomes which is, again, the first membrane-bound compartment to which miRNAs were found to be associated and function (Lee et al., 2009, Nat Cell Biol; Gibbins et al., 2009, Nat Cell Biol)

3. Similarly, the authors do not cover miRNAs trafficking between subcellular compartments, which is a very timely topic, and merely specify  in their conclusion that it is largely unknown  (line 424). The authors should at the very least mention Corradi et al., 2020, EMBO J. 

Specific comments: 

Concerning the presented text, while it is thorough, I would propose the following minor modifications. 

4. The section on nuclear miRNAs is the most developed. Yet, it would be of interest that the authors speculate more on the possible mechanisms of action through which miRNAs modulate transcription by binding to promoters (for instance see line 122-125) and how miRNAs activate enhancers (see line 149).

Figures: 

5. FIg.2B > accessibility instead of accesibility 

6. Fig.3 - leave more space between A. B. and C otherwise the panels appears too crowded

Round 2

Reviewer 2 Report

Accept in present form

Reviewer 3 Report

The authors have addressed adequately my comments.